# Mitochondrial Collapse Responsible for Chagasic and Post-Ischemic Heart Failure Is Reversed by Cell Therapy Under Different Transcriptomic Topologies

**DOI:** 10.3390/cimb47110940

**Published:** 2025-11-12

**Authors:** Dumitru A. Iacobas, Shavaiz Manzoor, Dennis Daniels, Sanda Iacobas, Lei Xi

**Affiliations:** 1Undergraduate Medical Academy, School of Public and Allied Health, Prairie View A&M University, Prairie View, TX 77446, USA; dedaniels@pvamu.edu; 2School of Medicine, Virginia Commonwealth University, Richmond, VA 23298, USA; manzoorsp2@vcu.edu; 3Department of Pathology, New York Medical College, Valhalla, NY 10565, USA; sandaiacobas@gmail.com; 4Division of Cardiology, Pauley Heart Center, Virginia Commonwealth University, Richmond, VA 23298, USA; lei.xi@vcuhealth.org

**Keywords:** cellular respiration, expression control, expression correlation, genomic fabric, NADH dehydrogenase, oxidative phosphorylation

## Abstract

Although experimental evidence indicates that mitochondrial collapse is a common effect of both Chagas disease and post-ischemic heart failure and that cardiac anatomy and function are partially restored by stem cell therapy, the responsible molecular mechanisms are still under debate. Gene expression data from our publicly accessible transcriptomic dataset obtained by profiling the left ventricle myocardia of mouse models of Chagas disease and post-ischemic heart failure were re-analyzed from the perspective of the Genomic Fabric Paradigm. In addition to the regulation of the gene expression levels, we determined the changes in the strength of the homeostatic control of transcript abundance and the remodeling of the gene networks responsible for the mitochondrial respiration. The analysis revealed that most of the mitochondrial genes assigned to the five complexes of the respiratory chain were significantly downregulated by both Chagas disease and ischemia but exhibited outstanding recovery of the normal expression levels following direct injection of bone-marrow-derived stem cells. However, instead of regaining the original expression control and gene networking, the treatment induced novel mitochondrial arrangements, suggesting that multiple transcriptomic topologies might be compatible with any given physiological or pathological state. This study confirmed several established mechanisms and identified novel gene expression signals, especially *Cox4i2, Cox6b1, Cox7b, Ndufb11*, and *Tmem186*, that warrant further investigations. Their broad rescue with cell therapy underscores mitochondria as a convergent, tractable target for cardiac repair.

## 1. Introduction

Mitochondria are critical organelles that regulate cell metabolism and survival, especially in the heart, where mitochondria comprise 30–40% of cardiomyocyte volume and play a primary role in energy production, calcium homeostasis, and regulation of cellular apoptosis. Mitochondrial dysfunction leads to impaired cardiomyocyte function, a hallmark of heart failure [1]. Since mitochondria are essential for the cellular energy-demanding cardiac contractile function, a decline in mitochondrial biogenesis and function [2] was associated with the development of ventricular dysfunction caused by myocardial infarction [3], type 2 diabetes [4], Chagas disease [5], or anticancer-drug-related cardiotoxicity [6] among several cardiovascular afflictions.

Decades after initial infection with the parasitic euglenoid *Trypanosoma cruzi* [7], transmitted by the so-called “kissing bug” [8], ~30% of individuals can develop chronic Chagas disease (CCC), a congestive heart failure and dilated cardiomyopathy [9,10,11]. It is estimated that CCC affects about 7 million people worldwide, most of them in Latin America [12], and it recently became endemic even in the United States [13]. Although the pathogenesis of CCC remains a matter of debate [14,15], the involvement of cardiac mitochondria was first demonstrated by Garg et al. in 2003 [16] and confirmed by Báez et al. in 2011 [17]. It is known that a year post-infection parasite persistence and inflammation are associated with structural and functional alterations in mouse cardiac mitochondria in a parasite-strain-dependent manner [18]. Common comorbidities include dyslipidemia and hypertension [19] and might lead to cryptogenic stroke [20]. As expected for any infectious disease, CCC triggers the immune response [21] that changes during the development of the disease [22].

Myocardial infarction [23], described as the cardiomyocytes’ death caused by insufficient oxygen supply, whose definition and management are still under debate [24,25], is directly related to mitochondrial dysfunction [26] and affects several functional pathways [27,28].

Previous studies on mouse models have shown that myocardial infarction induced by interruption of blood supply [29] activates a strong inflammatory response and leads to ventricular remodeling and ischemic heart failure (IHF) [30]. These effects were reversed by injecting bone-marrow-derived mononuclear stem cells into the cardiac scar tissue [31]. Whatever the cause, heart failure has severe consequences on all organs, including triggering various mental disorders [32].

Transcriptomic studies on cardiomyocytes and the left heart ventricle of CCC mice revealed remodeling of the immune response [33], extracellular matrix, cell adhesion [34], intercellular communication via gap junction channels [35,36], mitochondrial oxidative phosphorylation [37], JAK/STAT signaling, and cell cycle [38] functional pathways.

The progenitor cells are well recognized for their therapeutic potential in cardiac regeneration [39]. We found that injecting bone-marrow-derived stem cells restores most of the heart function and recovers most of the normal heart gene expression profile [40,41].

This research investigated the mitochondria-related genomic alterations in failing hearts due to CCC and IHF before and after stem cell treatment. We analyzed the KEGG-constructed oxidative phosphorylation functional pathway [42] as arranged in the mitochondrial module of the KEGG-constructed cardiovascular disease pathway of diabetic cardiomyopathy [43]. This study did not stop at the alteration of the gene expression profile but extended to encompass the entire mitochondrial transcriptomic topology, i.e., to include the control of transcripts’ abundances and the transcriptomic networks, as common in our Genomic Fabric Paradigm [41].

## 2. Materials and Methods

### 2.1. Experimental Data

We re-analyzed publicly accessible gene expression data generated in former Iacobas’ lab by profiling the transcriptomes of the left heart ventricle myocardium of adult, age-matched male C57Bl/6j mouse models of CCC [44] and IHF [45] before and after the treatment with bone-marrow-derived mononuclear cells [46,47]. The treated stem cells were collected from femurs and tibiae of adult C57Bl/6 mice and purified by centrifugation in a Ficoll gradient [30]. Chagas disease was induced by infecting adult male C57Bl/6j mice with trypomastigotes of the Colombian *Trypanosoma cruzi* strain [48], while the permanent myocardial infarcts were obtained by ligating the descending branch of the left coronary artery [39]. The experimental protocols and raw and normalized expression data are fully described and publicly accessible in the Gene Expression Omnibus (GEO) of the (USA) National Center for Biotechnology Information (NCBI) [44,45,46,47].

If in a microarray a spot had corrupted or saturated pixels, or the background exceeded half of the foreground fluorescence signal, that spot was eliminated from the analysis for all microarrays in the experiment. The background-subtracted foreground fluorescence signal from each spot was normalized to the median for that channel of the microarray block printed with the same pin and then averaged for all biological replicas of the corresponding condition. This “multiple yellow strategy” adopted by us in hundreds of transcriptomic experiments is theoretically the least affected by the bias between the two laser channels of the scanner and the potential differences among the spotting pins of the microarray printer.

For each cardiomyopathy, we examined two conditions: IN (infected/infarcted—not treated) and IT (infected/infarcted—treated) and compared them with CN (healthy/control—not treated), all in four biological replicates.

### 2.2. Primary Independent Characteristics of Individual Genes

Background-subtracted foreground fluorescence signals were averaged for spots redundantly probing the same gene. According to our standard procedure [41] (flowchart in Figure 1), each properly quantified gene was characterized in each condition by three independent characteristics: AVE, REV, and COR.

AVE is the average expression level across biological replicas. Comparing AVE values of a gene in two conditions indicates whether expression of that gene was altered.

REV (relative expression variation) is defined as the mid-interval of the chi-square estimate of the coefficient of variation of the expression levels on all spots redundantly probing the same gene across biological replicas of one condition. REV is used to calculate the REC (relative expression control) and RCS (relative control strength) as
(1)$$\forall c=CN,IN,IT\;,\;{REC}_{i}^{\left(c\right)}\equiv {log}_{2}\left(\frac{{\langle REV\rangle }^{\left(C\right)}}{{REV}_{i}^{\left(c\right)}}\right)\;,\;{RCS}_{i}^{\left(c\right)}\equiv \frac{{\langle REV\rangle }^{\left(C\right)}}{{REV}_{i}^{\left(c\right)}}$$ where 〈REV〉^(c)^ is the median REV of all genes quantified in the condition *c* = *CN*, *IN*, *IT*. A positive/negative REC value indicates that the expression of that gene is more/less controlled (i.e., more/less stably expressed across biological replicas) than the median in the condition. The control level of a gene indicates the importance of that gene for the cell survival and phenotypic expression, with the top positive value pointing to the most critical mitochondrial gene for the respective condition.

COR is the Pearson pairwise correlation coefficient of the (log_2_) expression levels of two genes. There are three (*p* < 0.05) statistically significant cases when each of the two genes “*i*” and “*j*” in condition “*c*” are probed by single microarray spots: (1)$1\geq {COR}_{ij}^{\left(c\right)}\geq 0.951$ synergistically expressed, i.e., their expression levels oscillate in phase across biological replicates.(2)$-1\geq {COR}_{ij}^{\left(c\right)}\leq -0.951$ antagonistically expressed, i.e., their expression levels oscillate in antiphase across biological replicates.(3)$\left|{COR}_{ij}^{\left(c\right)}\right|<0.05$ independently expressed genes, i.e., the expression of one gene has no influence on the expression of the other gene.

COR analysis was used to identify the (*p* < 0.05) significant transcriptomic networks based on the “principle of transcriptomic stoichiometry” [49], requiring coordinated expressions of genes whose encoded products are linked in a functional pathway. The overall expression correlation of the mitochondrial genes responsible for adjacent complexes (Cx) of the respiratory chain was measured by their coordination (COORD) percentage:
(2)$$\begin{array}{cc} \forall c=CN,IN,IT & \&\;\forall k=1÷4\;,\;COORD\left({Cx}^{k}\to {Cx}^{k+1}\right)\\ & \equiv SYN\left({Cx}^{k}\to {Cx}^{k+1}\right)+ANT\left({Cx}^{k}\to {Cx}^{k+1}\right)-IND\left({Cx}^{k}\to {Cx}^{k+1}\right) \end{array}$$ where SYN, ANT, and IND are the percentages of the *p* < 0.05 significant synergistically, antagonistically, and independently expressed gene pairs out of all pairs that can be formed between the two complexes.

REC and COR values were used to hierarchize the genes according to their Gene Commanding Height (CGH) score [50]:
(3)$$\forall c=CN,IN,IT\;,\;{GCH}_{i}^{\left(c\right)}\equiv exp\left({REC}_{i}^{\left(c\right)}+\langle {2{COR}_{iJ}^{\left(c\right)}}^{2}\rangle \right)$$

### 2.3. Transcriptomic Changes in Individual Genes and Functional Pathways

Our standard protocol [51] considers the expression of a gene as statistically significantly regulated (here in the left heart ventricle of the diseased mouse without and with cell treatment (d = IN, IT) with respect to the control counterpart CN) if the absolute expression fold-change
$\left|x_{i}^{\left(\frac{d}{CN}\right)}\right|$ exceeds the appropriate cut-off and the *p*-value of the heteroscedastic t-test of the means equality is less than 0.05. The absolute fold-change cutoff (CUT) is computed for each gene by considering both the expression variabilities in the compared conditions and the technical noises of the probing microarray spots in the two sets of biological replicas.
(4)$$\forall d=IN,IT\;,\;x_{i}^{\left(\frac{d}{CN}\right)}=\left\{\begin{array}{lll} \frac{{AVE}_{i}^{\left(d\right)}}{{AVE}_{i}^{\left(CN\right)}} & if & {AVE}_{i}^{\left(d\right)}\geq {AVE}_{i}^{\left(CN\right)}\\ -\frac{{AVE}_{i}^{\left(CN\right)}}{{AVE}_{i}^{\left(d\right)}} & if & {AVE}_{i}^{\left(d\right)}<{AVE}_{i}^{\left(CN\right)} \end{array}\right.$$
(5)$$\left|x_{i}^{\left(\frac{d}{CN}\right)}\right|>{CUT}_{i}^{\left(\frac{d}{CN}\right)}\equiv 1+\frac{1}{100}\sqrt{2\left({{REV}_{i}^{\left(d\right)}}^{2}+{{REV}_{i}^{\left(CN\right)}}^{2}\right)}\;\&\;p_{i}^{\left(\frac{d}{CN}\right)}<0.05$$

Contributions of individual genes to the transcriptomic alteration in condition “d” (CCC, IHF) with respect to their healthy counterparts were measured by their WIR (weighted individual (gene) regulation) score:
(6)$$\forall d=CCC,IHF\;,\;{WIR}_{i}^{\left(\frac{d}{CN}\right)}\equiv {AVE}_{i}^{\left(CN\right)}\left(\left|x_{i}^{\left(\frac{d}{CN}\right)}\right|-1\right)\left(1-p_{i}^{\left(\frac{d}{CN}\right)}\right)$$

The overall alteration of a functional pathway Φ in a particular condition (CCC or IHF) was measured by the median WIR of the pathway genes. A positive median WIR indicates that the total number of (significantly and not significantly) upregulated genes in the pathway exceeded the number of the downregulated genes, and vice versa.

The alteration of expression control in CCC and IHF mice without cell treatment with respect to the condition CN was measured by the fold-change (FC) defined as
(7)$$\forall d=IN,IT\;,\;{FC}_{I}^{\left(\frac{d}{CN}\right)}=\left\{\begin{array}{lll} \frac{{RCS}_{i}^{\left(d\right)}}{{RCS}_{i}^{\left(CN\right)}} & if & {RCS}_{i}^{\left(d\right)}\geq {RCS}_{i}^{\left(CN\right)}\\ -\frac{{RCS}_{i}^{\left(CN\right)}}{{RCS}_{i}^{\left(d\right)}} & if & {RCS}_{i}^{\left(d\right)}<{RCS}_{i}^{\left(CN\right)} \end{array}\right.$$

The fold-change alteration (negative for downregulation) of GCH in a treated (IT) or untreated (IN) Chagas and post-ischemic heart failure is computed as
(8)$$\forall d=IN,IT\;,\;GCH-{FC}_{I}^{\left(\frac{d}{CN}\right)}=\left\{\begin{array}{lll} \frac{{GCH}_{i}^{\left(d\right)}}{{GCH}_{i}^{\left(CN\right)}} & if & {GCH}_{i}^{\left(d\right)}\geq {GCH}_{i}^{\left(CN\right)}\\ -\frac{{GCH}_{i}^{\left(CN\right)}}{{GCH}_{i}^{\left(d\right)}} & if & {GCH}_{i}^{\left(d\right)}<{GCH}_{i}^{\left(CN\right)} \end{array}\right.$$

## 3. Results

### 3.1. Both Chagasic Disease and Post-Ischemic Heart Failure Are Characterized by Substantial Downregulation of Mitochondrial Genes

Figure 2 presents the mitochondrial genes that were significantly regulated (according to the composite criterion of the absolute fold-change and *p*-value (condition 5)) in Chagas disease and post-ischemic heart failure. Thus, in the untreated CCC, four of the five complexes contain only downregulated genes, with *Cx4* making the exception by having one upregulated gene (*Cox6a1*), although two other genes (*Cox5a*, *Cox6b1*) were downregulated. The percentage of downregulated genes is even larger in the post-ischemic heart despite the upregulation of *Ndufa9*.

The large numbers of the downregulated genes in the five complexes indicate a significant reduction in oxidative phosphorylation and, by consequence, less production of ATP. Remarkably, in both cardiomyopathies, Ca^2+^ storage in the mitochondrion was diminished by downregulation of one of the involved genes (*Vdac1* in CCC and *Slc25a5* in IHF mice). The downregulation of these transporters consistently led to Ca^2+^ accumulation in the cardiomyocyte that affected the contractility function of this heart-muscle cell.

### 3.2. Cell Treatment Restores the Normal Expression of Most Mitochondrial Genes Altered in Both Chagasic Disease and Post-Ischemic Heart Failure

As illustrated in the Figure 3, we found that the expressions of most mitochondrial genes that have been altered in both Chagasic and ischemic mice were markedly restored following the cell treatment.

Moreover, in the case of CCC mice, the initially not affected Ndufb11 was upregulated by treatment, while the originally downregulated Ndufs4 was even switched to upregulated. Of note is that normal Ca^2+^ storage in the mitochondrion was restored in the treated CCC mice but remained diminished in the treated IHF now because of the downregulation of *Vdac2* despite recovering the normal expression of initially downregulated *Slc25a5*.

### 3.3. The Largest Mitochondrial Gene Contributors to the Transcriptomic Alterations in Both Chagasic and Post-Ischemic Mice

Figure 2 and Figure 3 implicitly consider significantly regulated genes as equal contributors to the transcriptomic alteration, presenting the overall change as percentages of up- and downregulated genes, which is an oversimplified description of the biological reality. A better representation of the contribution would be by the pair (expression ratio, *p*-value) of each affected gene. However, we think that a more comprehensive characterization is by the weighted individual (gene) regulation (WIR), computed with Formula (6), that incorporates the total expression change AVE^(CN)^(|x| − 1) and the statistical confidence (1 − *p*-value) of this change.

Table 1 presents the largest five contributors (as absolute value of WIR) in Chagasic (CCC) and ischemic (IHF) mice, together with their normal average expression levels (AVE), expression ratios (X), and *p*-values of their regulation with respect to the normal expression. As expected, all top contributors had negative expression ratios.

The effectiveness of the cell treatment in restoring the expression profiles of the mitochondrial genes is evident by the change in the median WIR from −18.74 in infected not treated to −4.50 after treatment in CCC mice and from −16.83 to −1.81 in IHF mice. These results indicate both substantial reduction in mitochondrial function caused by parasitic infection and partial recovery following cell treatment. While in CCC mice, the non-mitochondrial gene *Pln* was the largest contributor (WIR = − 222), in the ischemic mice, one mitochondrial gene, *Cox5a*, tops the list with the impressive WIR = −2603(!).

Interestingly, the median WIR value for the entire transcriptome was positive in all conditions, meaning that overall, the contributions of the upregulated genes (like those involved in the inflammatory response as previously reported [23,39]) exceeded those of the downregulated. As expected, the median WIRs of the entire transcriptome were larger for both untreated CCC (0.67 vs. 0.13) and IHF (1.63 vs. 0.70) mice, pointing to an overall reduction in the alteration of the expression profile.

Figure 4 is a graphical representation of the fold-change (negative for downregulation) with respect to the healthy condition of the five largest mitochondrial gene contributors to the transcriptomic alterations in the two treated and untreated cardiomyopathies studied (data from Table 1).

### 3.4. The Most and the Least Controlled Mitochondrial Genes in CCC and IHF Mice

The experimental design with four biological replicates allowed us to estimate the relative expression control (REC) of individual genes in each condition and how much both treated and untreated cardiomyopathies affected them. Table 2 presents the REC values (computed according to Equation (1)) of the most (REC > 0) and the least (REC < 0) controlled mitochondrial genes in all profiled conditions (healthy, treated, and untreated CCC and IHF mice). We also listed the most and the least controlled genes in the entire transcriptome and the fold-change (negative for downregulation) of diseased states with respect to the healthy mice.

### 3.5. Both CCC and IHF Alter the Transcriptomic Networks of the Mitochondrial Genes by Partially Decoupling the Oxidative Phosphorylation Complexes

Figure 5 presents the (*p* < 0.05) significant transcriptomic networks coupling the five complexes of the oxidative phosphorylation in the left heart ventricles of healthy and untreated CCC and IHF mice. One observes that the adjacent complexes are coupled in all conditions mostly by synergistic expression correlations and that the coordination degree (computed with Formula (2)) is significantly lower in the untreated cardiomyopathies, indicating a partial decoupling of the complexes. The coordinated percentages were computed for the 80 possible gene pairs between Cx1 and Cx2, 27 between Cx2 and Cx3, 84 between Cx3 and Cx4, and 140 between Cx4 and Cx5.

The reduction was much more considerable for the untreated IHF mice, going to even negative values (COORD^(Cx1-Cx2)^ = −2.50%, COORD^(Cx3-Cx4)^ = −1.19%), which means more significantly independently that synergistically + antagonistically expressed gene pairs were identified. For instance, the number of synergistically + antagonistically expressed gene pairs between complexes Cx1 and Cx2 decreased from 18 (=18 + 0) in CN to 15 (=15 + 0) in CCC and to 6 (4 + 2) in IHF, while the number of independently expressed pairs goes from five in CN to four in CCC and eight in IHF. In general, there are fewer antagonistically than independently expressed gene pairs, explaining the synchrony of the respiratory chain.

### 3.6. Stem Cell Treatment Benefits for the Transcriptomic Coupling of the Oxidative Phosphorylation Complexes

Figure 6 presents the (*p* < 0.05) significant gene expression correlations in treated CCC and IHF mice and the coordination degrees in all five conditions. As summarized in Figure 6c, both cardiomyopathies reduced the expression coordination degrees among all pairs of coupled complexes. However, the stem cell treatment partially restored the coordination degree in the infarcted mice while even substantially increasing it in the CCC mice.

One important observation is that although the cell treatment restored the normal expression levels for most mitochondrial genes, the transcriptomic coupling of the complexes exhibits different than normal patterns. Importantly, the patterns of treated CCC and IHF are also largely distinct from one another.

### 3.7. Both CCC and IHF Alter the Hierarchy of Mitochondrial Genes

Table 3 presents the most prominent five mitochondrial genes in treated and untreated Chagasic and ischemic mice according to their Gene Commanding Height (GCH) scores computed with Formula (3). It also contains the fold-change (GCH-FC, negative for downregulation) of the GCHs in disease states with respect to the healthy condition, computed with Formula (8). For comparison, the table also presents the GCH scores of the top five mitochondrial genes in healthy mice and their GCHs in the untreated CCC and IHF mice.

## 4. Discussion

Alteration of the mitochondrial respiration and excessive generation of reactive oxygen species (ROS) are regarded as the main causes of heart failure [52,53], given that mitochondria provide about 90% of the energy used by the heart during normal functioning [54,55].

Using the Genomic Fabric Paradigm [51], which provides the most theoretically possible comprehensive transcriptomic characterization, our present study has brought new insights into the current understanding of mitochondrial gene involvement underlying Chagas disease [56] and post-ischemic heart failure [57]. Although several previous studies specifically analyzed alteration of the expression of genes/proteins that regulate mitochondrial oxidative phosphorylation [16,17], none of them studied the alteration and recovery of the gene expression control and transcriptomic networking.

Our transcriptomic characterization of untreated CCC and IHF hearts indicated a unified failure of mitochondrial energetics that was substantially reversed by bone-marrow-derived cell therapy. At the level of Complex 1 [58], the downregulation of 11/32 Nduf genes in untreated CCC and 22/32 in untreated IHF mice likely predicts a severe block in NADH oxidation, a surge in reactive oxygen species, and consequent energy depletion. Complex 1 (NADH ubiquinone oxidoreductase [59]) is responsible for NADH oxidation and proton pumping along the electron transport chain [60]. For instance, the downregulation of *Ndufa7* found by us (x = −1.92 in the untreated CCC and x = −1.72 in untreated IHF) was also reported in cardiac hypertrophy [61]. Although *Ndufa10*, known for its role in spontaneous hypertension [62,63], was equally downregulated (x = −1.60) in both untreated cardiomyopathies, its downregulation was statistically significant only in IHF mice (p^(IHF)^ = 0.009), while in untreated CCC mice it was not because of larger biological variability (p^(CCC)^ = 0.142). Interestingly, while *Ndufa9* was (even if not statistically significant) downregulated in untreated CCC, it was upregulated (x = 4.18) in untreated IHF, as happens in high-intensity training [62].

The collapse of 2/3 of the quantified Complex 2 genes, critical for the Krebs cycle [64,65,66,67], in both untreated heart diseases investigated by us, resulted from the significant downregulation of *Sdha* (x^(CCC)^ = −2.49, x^(IHF)^ = −2.29) and *Sdhb* (x^(CCC)^ = −2.07, x^(IHF)^ = −1.59).

Electron flow stops at Complex 3: the hinge protein *Uqcrch* shows one of the strongest negative WIR scores (WIR = −44.38), consistent with knockout data in which its loss alone causes contractile failure [68], thereby supporting and broadening those functional observations to two distinct heart failure models. Complex 3 (CoQ: cytochrome c-oxidoreductase [69]) deficiency is responsible for fewer mitochondrial disorders [70], although carbonylation of genes like *Uqcrc2* (found as downregulated in both untreated CCC and IHF) is a direct response to the stress of *Trypanosoma cruzi* infection [71].

Complex 4 is compromised by suppression of the ATP/ADP sensor *Cox4i1* in CCC, which matches the marked reduction in state-3 respiration and cytochrome-c-oxidase deficits clinically documented [72]. The downregulation (x = −2.3, WIR = −80) in untreated IHF mice of *Cox7b* likely undermines Complex 4 assembly, highlighting a novel candidate for future functional probing. *Cox5a*, the largest contributor listed in Table 1 to the downregulation of Cx4 in IHF (WIR = −2603), is considered a biomarker for the blood stasis syndrome [67], cyanotic heart disease [73], and acute myocardial infarction [74].

Adaptive changes in energy preservation were evident at Complex 5, where the ATP synthase ε-subunit *Atp5f1e* was downregulated (x = −2.22) in the ischemic hearts and then normalized after therapy. This not only agrees with but also now quantifies the prior enzymology-based findings [75,76]. Our results about the downregulation of Cx 5 contradict the report of its upregulation found by another group [77] through meta-analysis of 13 patients with heart failure subjected to heart transplantation compared to 10 healthy counterparts. However, Cpt2 (Carnitine palmitoyltransferase 2), not affected by either CCC or IHF, is downregulated in treated IHF, which, according to some research [78], might result in rapamycin-resistant, acetylation-independent hypertrophy.

By comparing Figure 2 and Figure 3, one observes that the cell treatment triggered marked recovery of the expression levels of almost all mitochondrial genes in both CCC and IHF mice. The recovery happens in all five oxidative phosphorylation complexes. While the commonly decreased expression of *Ndufs1* in the infarcted heart [79] is restored to the normal level by the cell treatment in our model, it remains (even less but still) downregulated in the treated CCC mice (from x^(IN)^ = −3.02 to x^(IT)^ = −2.20). However, the downregulation of *Ndufs4* in *T. cruzi*-infected mice was reversed to upregulation by the cell treatment, confirming the cardioprotection role of its upregulation [80].

A very important part of this study was devoted to the control of the transcripts’ abundances by the cellular homeostatic mechanisms and its modification during the disease development and following the treatment. Among the most controlled genes (Table 2), there were five subunits of NADH:ubiquinone oxidoreductase (i.e., mitochondrial complex I), including *Ndufa10*, *Ndufb10*, *Ndufb5*, *Ndufb11*, and *Ndufaf4*. These genes can affect mitochondria function profoundly. Especially *Ndufa10*, the mitochondrial gene with the largest REC (relative expression control) in healthy mice whose RCS (relative control strength) was markedly reduced in both treated (by 32.80×) and untreated (by 22.70×) CCC, meaning a high expression flexibility to adapt to changeable environmental conditions that might eventually be used to alleviate diabetic cardiomyopathy [81]. *Ndufa10* has also been implicated in diabetic cardiomyopathy [81] and ischemic injury [72]. Note also the substantial reduction in the expression control in both treated and untreated CCC mice for other stably expressed mitochondrial genes in the healthy mice: *Cox7b* and *Ndufb10*. To our best knowledge, this is the first report revealing the significant alteration of the expression control of these genes in Chagasic hearts. *Ndufb10* observed destabilization agrees with its known role in holoenzyme assembly failure [82].

The high RCS of *Tmem186* in the entire normal transcriptome was significantly reduced (RCS-FC = −39.13) in untreated IHF that persisted (RCS-FC = −22.50) after treatment, but the role of this gene is much less understood. *Tmem186* is a component of the mitochondrial Complex 1 intermediate assembly [58]. In contrast, IHF increased by 36.56× the control strength of the antigen *Cd164*, but the cell treatment restores the normal transcription control.

There are several other interesting data in Table 2. For instance, while the RCS of *Vdac2*, whose downregulation is one of the main responsible factors for heart failure [83], increased by 2.89× in the not-treated CCC mice (IN) with respect to the healthy animals (CN), it decreased in treated mice (IT) by 3.23×. The control of *Cox6b1*, known to relieve hypoxia/reoxygenation injury [83] increased by 8.31× in the untreated IHF but decreased by 2.48 in treated IHF. The situation is opposite for *Ndufaf4*, whose expression control decreased in untreated CCC but increased in treated CCC. We found substantial changes in ischemic mice, with some genes exhibiting more expression control (e.g., *Atp5f1e*, *Cox6b1*). The control of *Atp13a2* decreased in untreated IHF to considerably increase (by 14.82×) in treated IHF.

All these differences indicate that the homeostatic control mechanisms to keep the expression levels within certain intervals are neither uniform among genes nor similarly altered by each of the two cardiomyopathies nor restored following the same therapeutic approach. A very important observation comparing data from Table 2 with Figure 2 and Figure 3 is that, even though the cell treatment restored the normal expression of several genes, their expression control was not. For instance, in the ischemic mice, the expression of the significantly downregulated *Cox6b1* (x^(IN)^ = −2.87) was restored (x^(IT)^ = 1.1) by the treatment, but its RCS was decreased by 2.48×, while the recovered downregulation of *Atp5f1e* (x^(IN)^ = −2.22) was accompanied by an increase in the RCS by 2.77×). This observation indicates that whereas the treatment recovered the expression profile of most mitochondrial genes, it did not restore the initial control of the transcripts’ abundances.

Recent work in acute experimental models of Chagas disease showed that an imbalance between ROS and NO creates a strongly proarrhythmic substrate independent of major structural remodeling and directly links mitochondrial redox output to electrical instability [2,84]. The same study demonstrates that NO inhibition reverses early after-depolarizations and action potential alternans, underscoring redox–calcium crosstalk as a therapeutic lever. Interestingly, the parasite developed a system of antioxidant enzymes by which to resist the high ROS concentration in the infected cardiomyocyte [85]. Our findings of Complexes 1–5 suppression and *Vdac1* dysregulation align with this mechanism and suggest that transcriptomic loss of the ETC (electron transfer chain) control may be an upstream driver of the ROS/NO disequilibrium [86]. There is no contradiction between our finding of downregulation of *Slc25a5*, one of the genes responsible for pumping in Ca^2+^, and recent reports of mitochondrial Ca^2+^ overload during the acute phase of the myocardial ischemia-reperfusion [87] because in the post-ischemic failing heart (our case), mitochondrial Ca^2+^ content is lower [88].

Additional analysis highlighted that mitochondrial injury is shaped by both inflammatory environment and genotype [89]. This supports our observation that nuclear-encoded ETC subunits (*Ndufb11*, *Cox7b*) are vulnerable nodes that could synergize with inflammatory signaling to depress ATP output. It also strengthens the rationale to stratify future cohorts by mitochondrial genotype and inflammatory profile when validating control metrics [81,90]. There was evidence for adipose tissue as a chronic reservoir for *Trypanosoma cruzi* and a source of adipokine-associated mitochondrial stress, implying that systemic metabolic inputs can burden cardiac bioenergetics [86]. This concept aligns with our *Cpt2* signal finding that reduced fatty acid oxidation capacity in the heart may be compounded by parasite-driven adipose dysfunction and argues for integrating peripheral metabolic tissues into future mechanistic and therapeutic designs [91]. Moreover, the downregulation of the two pyruvate dehydrogenase modules (PDK and PDH) generating lipotoxicity by reducing the oxidation of the fatty acids [92] in both untreated cardiomyopathies was corrected by the cell treatment (Figure 3).

Figure 5 presents for the first time to our knowledge the transcriptomic networks that couple the respiratory chain complexes. It also shows how infection with *Trypanosoma cruzi* and ischemia alter the transcriptomic coupling, indicating profound remodeling of the oxidative phosphorylation functional pathway. The most dramatic alteration occurred between Cx1 and Cx2 in the IHF mice, where the coordination degree decreased from +16.25% to −2.50%, meaning a significant decoupling of the paired genes between the two complexes. The marked coupling decrease between the adjacent complexes is an additional explanation of the mitochondrial failure in the two investigated cardiomyopathies.

Understanding the importance of some of these expression correlations for normal heart function and what pathological consequences their alterations bring might open a very interesting and challenging field of research. For instance, Chagas disease turned the significant synergism of the downregulated *Nfdufa6* and *Sdha*, whose simultaneous upregulation is an indication of acute myeloid leukemia [93], into statistically significant independence. We found that the pairs *Uqcrb* and *Coxcb*, and *Atp5pf* and *Cox5b*, reported as simultaneously upregulated in paradoxical low transvalvular flow and low gradient patients who develop advanced heart failure symptoms [94], are synergistically expressed in normal conditions. The synergism of *Sdhb* and *Uqcrffs1* downregulated by ischemia supports the efforts to improve the diagnosis and gene-targeted therapies of Alzheimer’s disease [95].

It will be very interesting (and the object of a future study of our group) to investigate the interaction between the oxidative phosphorylation pathway and the mTOR pathway to understand their implication in the observed mitophagy in *T. cruzi*-infected cardiomyocytes [96]. It will also be of interest to study the interplay between the oxidative phosphorylation pathway and the PINK1-PRKN-mediated mitophagy [97]. Interesting future research should also investigate how Chagas disease and myocardial infarction affect the mitochondrial network plasticity [98].

As illustrated in Figure 6c, cell therapy recovered part of the decrease in the four coordination degrees in the IHF mice and even increased them over the original levels in the CCC mice. In the CCC mice, the cell therapy not only restored the normal expressions of the downregulated *Ndufv1* and *Sdhb* (also downregulated in diabetic patients caused by a high-fat diet [99]) but also made significant their synergistic expression. To prove this assertion, one can use an online available calculator [100] to determine the *p*-values of the statistical significance of the expression correlations in each of the three conditions: COR^(CN)^ = 0.832 (*p*-value = 0.168), COR^(IN)^ = 0.397 (*p*-value = 0.603), and COR^(IT)^ = 0.996 (*p*-value = 0.004). Interestingly, *Ndufa13* and *Sdhd*, which are independently expressed in healthy mice housed in normal atmospheric conditions, maintained their independence in untreated IHF but became synergistically expressed in both treated CCC and IHF mice. The treatment turned the practically neutral correlations: COR^(CN)^ = 0.320 (*p*-value = 0.680) and COR^(IN)^ = 0.222 (0.778) between the downregulated *Cox17* and *Atp5mc1* in ischemic mice into a significant antagonistic expression: COR^(IT)^ = −0.952 (*p*-value = 0.048). However, together with *NDUFB1*, *COX17*, and *ATP5MC1*, they were reported as upregulated in patients with Parkinson’s, Alzheimer’s, and Huntington’s diseases [101].

Nevertheless, we must recognize that the correlation analysis cannot determine which gene of the pair is the master, i.e., whose gene expression level commands the expression level of the other, nor about the energy coupling [102] and the electric charge transfer within or between the respiratory complexes [103,104,105].

The prominence analysis ranked the mitochondrial genes according to their importance for the functioning of the cell powerhouse. From Table 3 one may learn the following:(i)Gene hierarchy is altered in cardiomyopathy, as revealed by the differences between the top five genes in healthy mice compared to the sets of five in the other heart conditions.(ii)Each type of cardiomyopathy induces distinct alteration of the gene hierarchy (see GCH differences between CCC and IHF for the most prominent genes in healthy mice).(iii)Restoration of the normal expression level is not accompanied by the reinstatement of the genes in their right hierarchy, indicated by the non-unit GCH-FC. For instance, *Uqcrh*, found by us as downregulated in both untreated CCC and IHF mice, was reported as upregulated in patients with hypertrophic cardiomyopathy [106]. Cell treatment restored the normal expression in the CCC but not in the IHF mice. However, the restored expression level is not followed by the full restoration of its GCH after the treatment. Instead, its RCS became 6.67× stronger than in the normal condition (even stronger than in the untreated CCC, where it was 4.66×).(iv)Cell treatment has different effects on the two types of heart afflictions. For instance, the somehow low GCH of *Cox6b1* healthy mice that was considerably raised in both untreated cardiomyopathies (GCH-FC^(IN-CCC)^ = 4.81, GCH-FC^(IN-IHF)^ = 5.53) is raised even more in treated CCC (GCH-FC) = 7.34) but downgraded in treated IHF (GCH-FC = −2.10).

In previous publications, we have shown that the most prominent gene of a condition might be the most effective target for the gene therapy of that condition [50,107]. Therefore, this analysis may open new therapeutic avenues for both ischemic failure and Chagas disease.

There are several reports from our former collaborators about the beneficial effects of stem cell therapy on the same mouse model of Chagas disease (e.g., [108,109]) in recovering the normal cardiac anatomy and functions that correlate with recovery of the gene expression profile. Also, our previous report on transcriptomic recovery of cell-treated IHF mice [31] mentioned that after 25 days of therapy, the mice demonstrated absence of the pathological Q wave in the electrocardiogram, improved systolic performance, and much less ventricular dilatation. However, a quantitative gene expression study like the present one cannot explain the complex interaction between the damaged cardiomyocytes and the stem cells [110,111] or the molecular mechanisms involved in the rescue of the mitochondrial gene expression profile.

## 5. Conclusions and Future Directions

Taken together, the findings of this study establish an exhaustive profile of mitochondrial collapse in Chagas disease and post-ischemic heart failure. This investigation demonstrated that targeted cell therapy can partially restore the functional integrity of key ETC components. It further confirmed several established mechanisms and introduced new signals. Their broad rescue with cell therapy underscores mitochondria as a convergent, tractable target for cardiac repair.

Heart malfunction affects all organs, brain included, and myocardial infarction is often followed by depression and other mental disorders [32]. Therefore, one major task of clinical management of the post-infarction mental problems is the restoration of the normal cardiac function. As such, cell treatment might be a solution, and clinical trials are underway [112,113].

Interestingly, although cell treatment recovers most of the normal gene expression profile, it does not restore either the normal expression control or the inter-complex transcriptomic coupling. Moreover, both treated cardiomyopathies present different expression control and correlation patterns. Since the treated animals displayed almost normal electrophysiological parameters and excitatory and contractility properties of the ventricular myocardium, our results suggest that a pathophysiological state is compatible with several transcriptomic topologies. The transcriptomic topology was mathematically defined by us ([114], Appendix A) as the weighted superposition of virtual transcriptomes where the expressions of the genes are controlled and correlated in groups of two, three, four, …. Given the stochastic nature of the gene transcription, transcript abundances can fluctuate within intervals subject to relative control strength. Therefore, each pathophysiological condition might be associated with a family of transcriptomic topologies, each defined by a distinct configuration function. This conclusion points to the need of defining a kind of “transcriptomic entropy of a state” that might be useful to assess the “transcriptomic health” of that state.

## Figures and Tables

**Figure 1 cimb-47-00940-f001:**
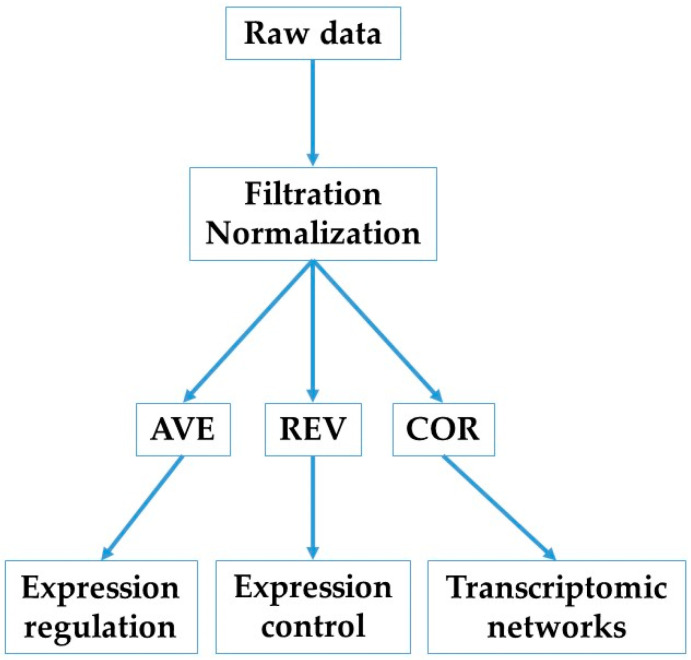
Flowchart of this study.

**Figure 2 cimb-47-00940-f002:**
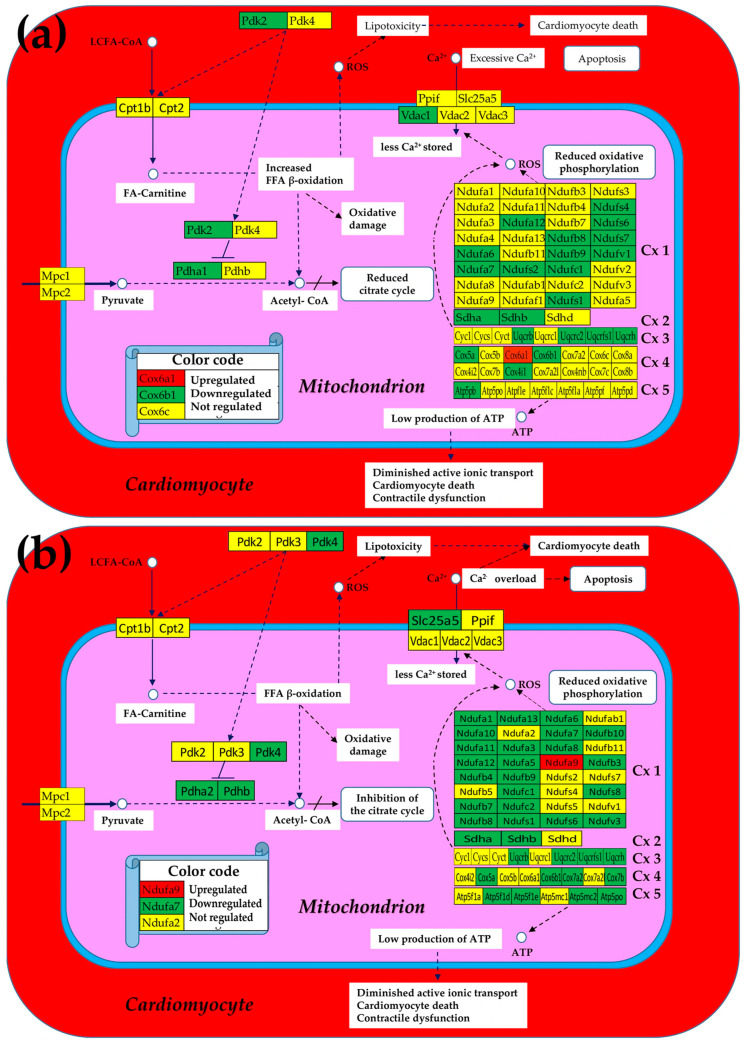
Significantly regulated mitochondrial genes in (**a**) Chagas disease and in (**b**) post-ischemic heart failure. A red/green background of the gene symbol indicates significant up/downregulation, while a yellow background indicates that the expression change was not statistically significant. Regulated genes: *Atp5f1d/e* (ATP synthase, H+ transporting, mitochondrial F1 complex, delta/epsilon subunit), *Atp5mc2* (ATP synthase membrane subunit c locus 2), *Atp5pb/o* (ATP synthase peripheral Uqcrh stalk-membrane subunit b/OSCP), *Cox4i1* (cytochrome c oxidase subunit IV isoform 1), *Cox17* (cytochrome c oxidase, subunit XVII assembly protein homolog (yeast)), *Cox5a* (cytochrome c oxidase, subunit Va), *Cox6a1/6b1* (cytochrome c oxidase, subunit VI a/b, polypeptide 1), *Cox7a2/b* (cytochrome c oxidase, subunit VIIa 2/VIIb), *Cpt2* (carnitine palmitoyltransferase 2), *Cyc1* (cytochrome c-1), *Ndufa6/7/12* (NADH:ubiquinone oxidoreductase subunit A6/7/12), *Ndufb8/9/10* (NADH: ubiquinone oxidoreductase subunit beta8/9/10), *Ndufc1/2* (NADH:ubiquinone oxidoreductase subunit C1/C2), *Ndufs1/4/6/7/8* (NADH dehydrogenase (ubiquinone) Fe-S protein 1/4/6/7/8), *Ndufv3* (NADH:ubiquinone oxidoreductase core subunit V3), *Pdha1* (pyruvate dehydrogenase E1 alpha 1), *Pdk2/4* (pyruvate dehydrogenase kinase, isoenzyme 2/4), Sdha/b (succinate dehydrogenase complex, subunit A/B), *Slc25a5* (solute carrier family 25 (mitochondrial carrier, adenine nucleotide translocator), member 5), *Uqcrc2* (ubiquinol-cytochrome c reductase core protein 2), *Uqcrfs1* (ubiquinol-cytochrome c reductase, Rieske iron–sulfur polypeptide 1), *Uqcrh* (ubiquinol-cytochrome c reductase hinge protein), *Uqcrb/h* (ubiquinol-cytochrome c reductase binding/hinge protein), and *Vdac1* (voltage-dependent anion channel 1).

**Figure 3 cimb-47-00940-f003:**
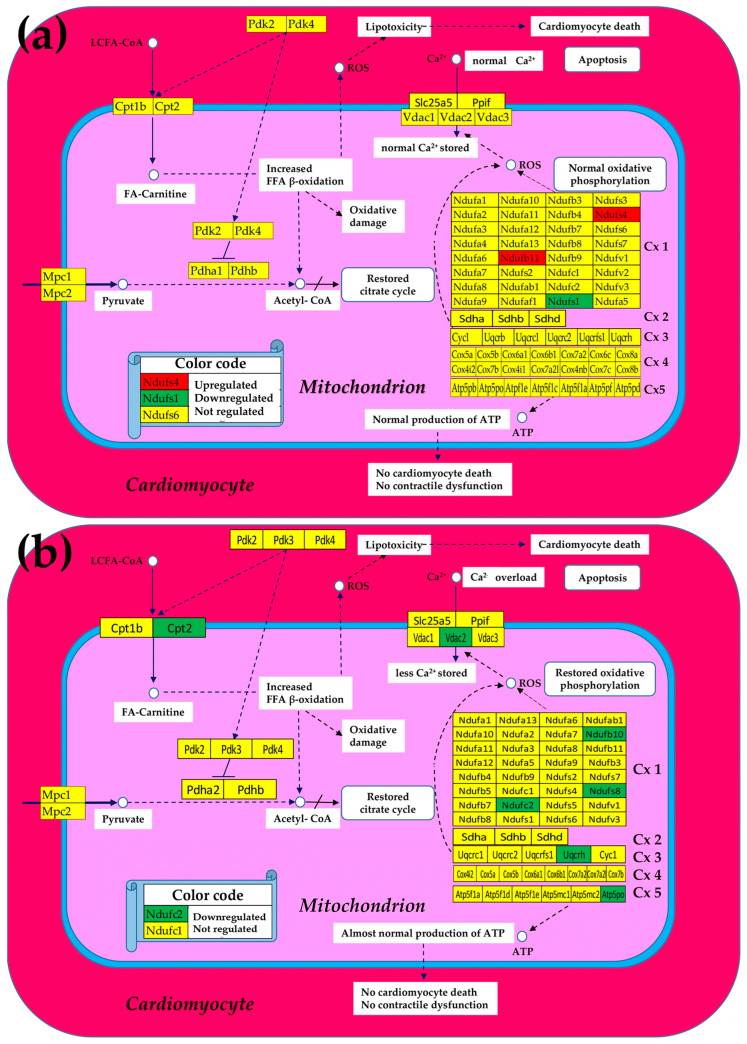
Remaining significantly regulated mitochondrial genes after stem cell treatment in (**a**) Chagas disease cardiomyopathy and in (**b**) post-ischemic heart failure. A red/green background of the gene symbol indicates significant up/downregulation, while a yellow background indicates that the expression change was not statistically significant. Regulated genes: *Atp5po* (ATP synthase peripheral *Uqcrh* stalk-membrane subunit OSCP), *Cpt2* (carnitine palmitoyltransferase 2), *Ndufb1/10/11* (NADH: ubiquinone oxidoreductase subunit beta1/10/11), *Ndufc2* (NADH:ubiquinone oxidoreductase subunit C2), *Ndufs1/4/8* (NADH dehydrogenase (ubiquinone) Fe-S protein 1/4/8), *Uqcrh* (ubiquinol-cytochrome c reductase hinge protein), and *Vdac2* (voltage-dependent anion channel 2).

**Figure 4 cimb-47-00940-f004:**
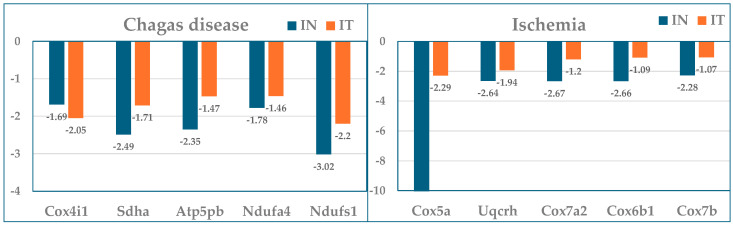
Expression ratios with respect to control (healthy mice) of the five largest mitochondrial gene contributors to the transcriptomic alterations in the treated (IT) and untreated (IN) Chagas disease and post-ischemic heart failure.

**Figure 5 cimb-47-00940-f005:**
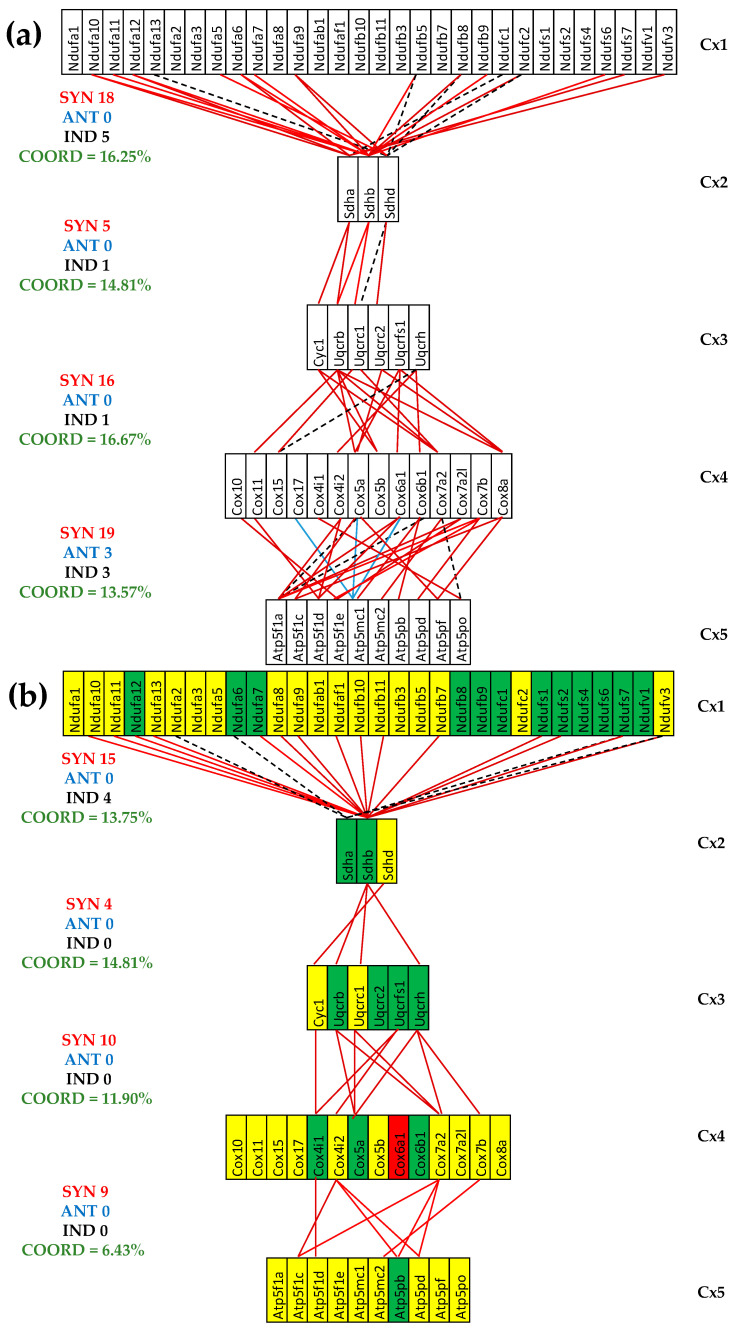
The (*p* < 0.05) significant transcriptomic networks coupling the five complexes of the oxidative phosphorylation in the left heart ventricles of healthy and untreated CCC and IHF mice. (**a**) Healthy mice, (**b**) untreated CCC mice, (**c**) untreated IHF mice. Continuous red/blue lines indicate (*p* < 0.05) significant synergistic/antagonistic expression correlations of the paired genes, while dashed black lines point to significantly independently expressed gene pairs. Missing lines mean that the expression correlation was not (*p* < 0.05) statistically significant. A red/green background of the gene symbols indicates significant up/downregulation, while a yellow background means no significant change in the expression level with respect to the healthy mice.

**Figure 6 cimb-47-00940-f006:**
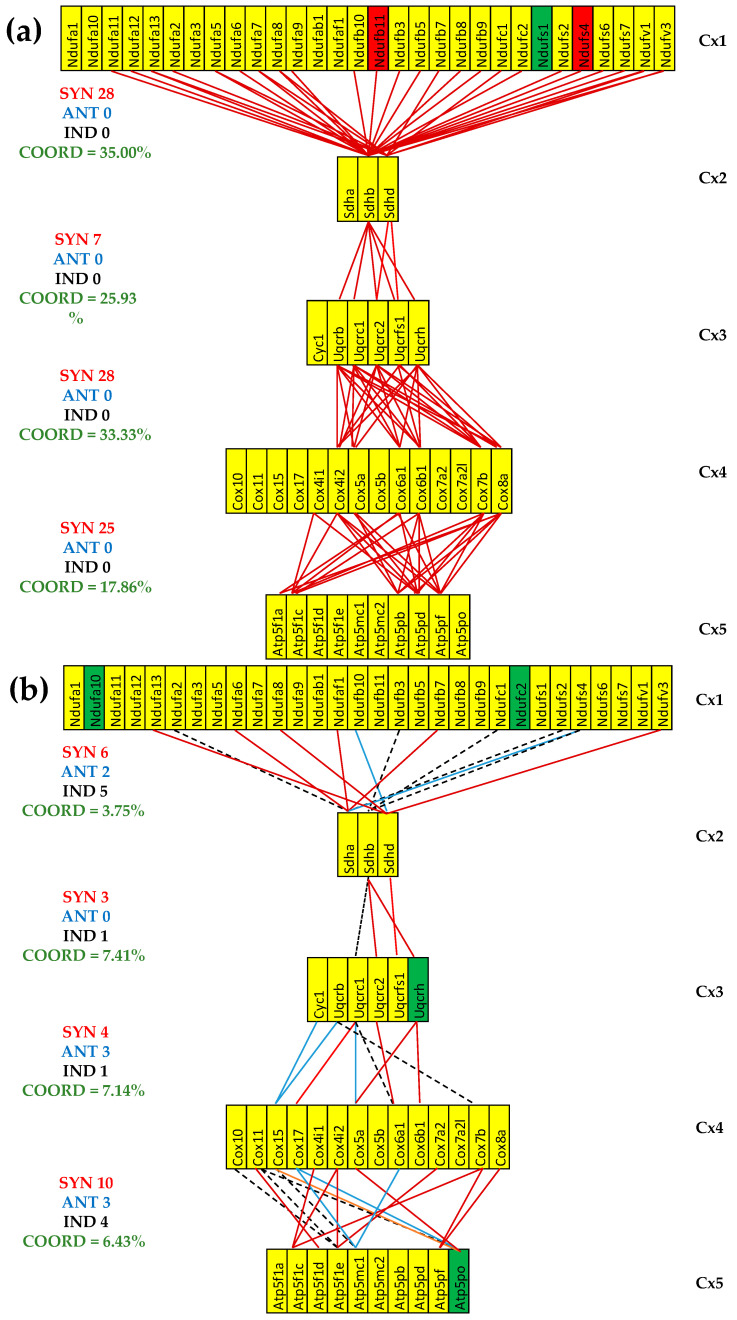
The (*p* < 0.05) significant transcriptomic networks coupling the five complexes of the oxidative phosphorylation in the left heart ventricles of treated CCC and IHF mice. (**a**) CCC-treated mice, (**b**) IHF-treated mice, (**c**) coordination degrees between adjacent complexes of the oxidative phosphorylation functional pathway in the left heart ventricle of healthy (denoted by CN) untreated CCC (CCI) and IHF (IHI) and treated CCC (CCT) and IHF (IHT) mice. Continuous red/blue lines indicate (*p* < 0.05) significant synergistic/antagonistic expression correlations of the paired genes, while dashed black lines point to independently expressed gene pairs. Missing lines indicate that the expression correlation of the two genes was not (*p* < 0.05) statistically significant. A red/green background of the gene symbols indicates significant up/downregulation, while a yellow background means no significant change in the expression level with respect to the healthy mice. Note the coordination reduction in both untreated cardiomyopathies, reaching even negative values for two pairs of complexes (Cx1-Cx2, Cx3-Cx4) in IHF mice. (**c**) IHF-treated mice.

**Table 1 cimb-47-00940-t001:** The largest mitochondrial (MITO) gene contributors to the heart transcriptome alteration in Chagasic (CCC) and ischemic (IHF) mice. AVE = average expression in the control mice, X-IN = expression ratio in untreated infected/infarcted with respect to control (negative for downregulation), X-IT = expression ratio in treated infected/infarcted with respect to control (negative for downregulation), |WIR| = absolute value of the weighted individual (gene) regulation. The gray background indicates the most important values in each condition.

**The Largest Contributors to the Mitochondrial Transcriptome Alteration in CCC Mice**								
**Gene**	**Description**	**AVE**	**X-IN**	* **p** *	**|WIR|**	**X-IT**	* **p** *	**|WIR|**
* **Cox4i1** *	**Cytochrome c oxidase subunit IV isoform 1**	165	−1.69	0.02	**111**	−2.05	0.45	95
*Sdha*	Succinate dehydrogenase complex, subunit A, flavoprotein	61	−2.49	0.00	91	−1.71	0.16	36
*Atp5pb*	ATP synthase peripheral stalk-membrane subunit b	62	−2.35	0.00	83	−1.47	0.27	21
*Ndufa4*	Mlrq-like protein	105	−1.78	0.09	76	−1.46	0.27	35
*Ndufs1*	NADH dehydrogenase (ubiquinone) Fe-S protein 1	34	−3.02	0.01	68	−2.20	0.05	39
	**Average mitochondrial |WIR|**				**54**			**49**
	**The largest overall contributors to the entire transcriptome alteration in CCC mice**							
*Pln*	Phospholamban	148	−2.64	0.09	**222**	−2.65	0.31	167
	**Overall average |WIR|**				**2.24**			**1.15**
**The Largest Contributors to the Mitochondrial Transcriptome Alteration in IHF Mice**								
**Gene**	**Description**	**AVE**	**X-IN**	* **p** *	**|WIR|**	**X-IT**	* **p** *	**|WIR|**
* **Cox5a** *	**Cytochrome c oxidase, subunit Va**	49	−59.69	0.09	**2603**	−2.29	0.28	45
*Uqcrh*	Ubiquinol-cytochrome c reductase hinge protein	59	−2.64	0.02	94	−1.94	0.03	54
*Cox7a2*	Cytochrome c oxidase, subunit VIIa 2	49	−2.67	0.00	81	−1.20	0.16	8
*Cox6b1*	Cytochrome c oxidase, subunit VIb polypeptide 1	49	−2.66	0.00	80	−1.09	0.48	2
*Cox7b*	Cytochrome c oxidase subunit VIIb	64	−2.28	0.03	80	−1.07	0.69	1
	**Average mitochondrial |WIR|**				**69**			**12**
	**The largest overall contributors to the entire transcriptome alteration in IHF mice**							
* **Cox5a** *	Cytochrome c oxidase, subunit Va	49	−59.69	0.09	**2603**	−2.29	0.28	45
	**Overall average |WIR|**				**1.63**			**0.70**

**Table 2 cimb-47-00940-t002:** The most and the least controlled mitochondrial genes in Chagasic and ischemic mice. Abbreviations: CN = control (reference); IN = infected/ischemic not treated; IT = infected/ischemic treated; MITO = mitochondrial; RCS-FC, fold-change of the relative control strength in IN/IT with respect to CN. For reference, the table includes the RECs and the RCS-FCs of the most and least controlled genes in the entire profiled transcriptome in control, untreated, and treated CCC mice. The gray background indicates the most important values in each condition.

**The Most and the Least Controlled Mito Genes in CCC Mice**					**RCS-FC**	
**Gene**	**Description**	**CN**	**IN**	**IT**	**IN**	**IT**
*Ndufa10*	NADH:ubiquinone oxidoreductase subunit A10	3.86	−1.17	−0.64	−32.80	−22.70
*Cox7b*	Cytochrome c oxidase subunit VIIb	2.59	−0.42	−1.59	−8.05	−18.12
*Ndufb10*	NADH:ubiquinone oxidoreductase subunit B10	2.15	−0.95	−0.68	−8.56	−7.13
*Vdac2*	Voltage-dependent anion channel 2	1.06	2.59	−0.63	2.89	−3.23
*Mpc2*	Mitochondrial pyruvate carrier 2	0.32	1.54	0.68	2.34	1.28
*Cox4nb*	COX4 neighbor	0.81	1.28	0.98	1.38	1.12
*Ndufb5*	NADH:ubiquinone oxidoreductase subunit B5	0.43	−0.53	1.66	−1.95	2.33
*Ndufb11*	NADH:ubiquinone oxidoreductase subunit B11	0.99	0.54	1.62	−1.36	1.55
*Ndufaf4*	NADH:ubiquinone oxidoreductase subunit A4	0.29	−0.92	1.37	−2.32	2.11
*Sdhd*	Succinate dehydrogenase complex, subunit D, integral membrane protein	−1.87	−0.32	−0.30	2.92	2.96
*Cox5a*	Cytochrome c oxidase, subunit Va	−1.58	1.17	−2.53	6.69	−1.94
*Ndufa9*	NADH:ubiquinone oxidoreductase subunit A9	−1.26	−2.64	−0.32	−2.60	1.93
*Cpt2*	Carnitine palmitoyltransferase 2	0.53	−1.67	0.21	−4.60	−1.25
*Pdk2*	Pyruvate dehydrogenase kinase, isoenzyme 2	0.60	−1.43	−0.67	−4.09	−2.41
*Ppif*	Peptidylprolyl isomerase F (cyclophilin F)	0.57	−1.41	−0.56	−3.94	−2.18
*Cox7b*	Cytochrome c oxidase subunit VIIb	2.59	−0.42	−1.59	−8.05	−18.12
*Atp5pf*	ATP synthase peripheral stalk subunit F6	1.52	−0.81	−1.58	−5.01	−8.53
*Cox4i1*	Cytochrome c oxidase subunit IV isoform 1	0.66	−0.20	−1.56	−1.81	−4.65
**The Most and the Least Controlled Mito Genes in IHF Mice**					**RCS-FC**	
*Cox6b1*	Cytochrome c oxidase, subunit VIb polypeptide 1	1.36	4.38	0.05	8.13	−2.48
*Atp5f1e*	ATP synthase F1 subunit epsilon	−0.79	2.68	0.68	11.07	2.77
*Atp5mc2*	ATP synthase membrane subunit c locus 2	0.52	1.75	−0.37	2.35	−1.86
*Ndufb11*	NADH dehydrogenase (ubiquinone) 1 beta subcomplex, 11	0.74	0.87	2.38	1.10	3.11
*Ndufv1*	NADH dehydrogenase (ubiquinone) flavoprotein 1	−1.14	−0.97	1.79	1.12	7.58
*Cpt1b*	Carnitine palmitoyltransferase 1b, muscle	0.16	−1.42	1.64	−2.99	2.79
*Ndufa9*	NADH:ubiquinone oxidoreductase subunit A9	−1.26	−2.64	−0.32	−2.60	1.93
*Cyc1*	Cytochrome c-1	−1.05	−1.67	0.93	−1.54	3.95
*Ndufa11*	NADH:ubiquinone oxidoreductase subunit A11	−0.36	−1.58	−0.29	−2.34	1.04
*Cox5a*	Cytochrome c oxidase, subunit Va	−1.58	1.17	−2.53	6.69	−1.94
*Mpc1*	Mitochondrial pyruvate carrier 1	−1.05	−0.92	−1.71	1.10	−1.58
*Ndufc2*	NADH:ubiquinone oxidoreductase subunit C2	0.38	−0.30	−1.52	−1.61	−3.75
**The most controlled genes in the entire transcriptome in CN and IHF mice**					**RCS-FC**	
*Tmem186*	Transmembrane protein 186	5.19	−0.10	0.70	−39.13	−22.50
*Cd164*	CD164 antigen	0.09	5.28	0.57	36.56	1.40
*Atp13a2*	ATPase type 13A2	0.66	−0.03	4.55	−1.61	14.82
**The least controlled genes in the entire transcriptome in CN and IHF mice**					**RCS-FC**	
*Gmcl1*	Germ cell-less homolog 1 (Drosophila)	−2.40	−0.33	−2.50	4.19	−1.07
*Idh3g*	Isocitrate dehydrogenase 3 (NAD+), gamma	−1.10	−2.83	0.80	−3.32	3.72
*Tsc22d4*	TSC22 domain family, member 4	0.98	0.25	−2.91	−1.67	−14.86

**Table 3 cimb-47-00940-t003:** The Gene Commanding Heights (GCHs) of the five most prominent mitochondrial genes in control, treated, and untreated Chagasic and ischemic mice. Abbreviations: IN = infected/ischemic not treated; IT = infected/ischemic treated; MITO = mitochondrial; GCH-FC = fold-change of the Gene Commanding Height; CCC = Chagasic mice; IHF = ischemic mice. For reference, the table includes the GCHs of the most prominent MITO genes in control mice and their scores in untreated CCC and IHF mice. The gray background indicates the most important values in each condition.

**Most Prominent Mito Genes in CCC Mice**		**GCH**		**GCH-FC**	
**GENE**	**Description**	**IN**	**IT**	**IN**	**IT**
* **Cox4i2** *	**cytochrome c oxidase subunit 4I2**	**13.20**	17.02	8.74	11.27
*Ndufb7*	NADH dehydrogenase (ubiquinone) 1 beta subcomplex, 7	11.73	17.82	14.09	21.42
*Cox6b1*	Cytochrome c oxidase, subunit VIb polypeptide 1	11.48	17.52	4.81	7.34
*Uqcrh*	Ubiquinol-cytochrome c reductase hinge protein	11.21	16.07	4.66	6.67
*Ndufs4*	NADH dehydrogenase (ubiquinone) Fe-S protein 4	11.00	12.23	4.66	5.18
* **Ndufa7** *	**NADH dehydrogenase (ubiquinone) 1 alpha subcomplex, 7**	7.48	**19.80**	1.42	3.77
*Ndufc1*	NADH:ubiquinone oxidoreductase subunit C1	4.60	19.71	−1.18	3.63
*Ndufa2*	NADH dehydrogenase (ubiquinone) 1 alpha subcomplex, 2	6.54	19.70	7.64	23.01
*Ndufa10*	NADH dehydrogenase (ubiquinone) 1 alpha subcomplex 10	9.54	19.57	1.71	3.52
*Uqcrb*	Ubiquinol-cytochrome c reductase binding protein	9.95	19.55	2.44	4.80
**Most Prominent Mito Genes in IHF Mice**		**GCH**		**GCH-FC**	
* **Cox6b1** *	**Cytochrome c oxidase, subunit VIb polypeptide 1**	**24.16**	2.08	5.53	−2.10
*Atp5f1e*	ATP synthase F1 subunit epsilon	9.01	2.00	3.70	−1.22
*Atp5mc2*	ATP synthase membrane subunit c locus 2	5.13	1.12	1.88	−2.44
*Ndufs5*	NADH dehydrogenase (ubiquinone) Fe-S protein 5	4.22	1.40	1.13	−2.65
*Ndufa1*	NADH dehydrogenase (ubiquinone) 1 alpha subcomplex, 1	4.21	3.60	1.38	1.18
* **Ndufb11** *	**NADH dehydrogenase (ubiquinone) 1 beta subcomplex, 11**	3.01	**6.39**	−1.90	1.12
*Ndufs4*	NADH dehydrogenase (ubiquinone) Fe-S protein 4	0.80	5.92	−9.72	−1.31
*Ndufv1*	NADH dehydrogenase (ubiquinone) flavoprotein 1	0.72	4.48	−3.68	1.68
*Cyc1*	Cytochrome c-1	0.52	4.26	−4.69	1.73
*Ndufa1*	NADH dehydrogenase (ubiquinone) 1 alpha subcomplex, 1	4.21	3.60	1.38	1.18
**Most Prominent Mito Genes in Healthy Mice**		**GCH**		**CCC**	**IHF**
* **Ndufb10** *	**NADH dehydrogenase (ubiquinone) 1 beta subcomplex, 10**	**13.07**		7.21	2.26
*Ndufa10*	NADH dehydrogenase (ubiquinone) 1 alpha subcomplex 10	9.79		9.54	1.01
*Uqcrb*	Ubiquinol-cytochrome c reductase binding protein	9.69		9.95	2.91
*Ndufaf1*	NADH:ubiquinone oxidoreductase complex assembly factor 1	8.14		8.31	1.67
*Ndufs4*	NADH dehydrogenase (ubiquinone) Fe-S protein 4	7.76		11.00	0.80

## Data Availability

Microarray protocol and raw data can be found in the databases https://www.ncbi.nlm.nih.gov/geo/query/acc.cgi?acc=GSE17363, https://www.ncbi.nlm.nih.gov/geo/query/acc.cgi?acc=GSE18703, https://www.ncbi.nlm.nih.gov/geo/query/acc.cgi?acc=GSE24088, https://www.ncbi.nlm.nih.gov/geo/query/acc.cgi?acc=GSE29769 (accessed on 15 September 2025).

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
