# Peer review of "Mitochondrial Collapse Responsible for Chagasic and Post-Ischemic Heart Failure Is Reversed by Cell Therapy Under Different Transcriptomic Topologies"

_cimb, 2025, doi:10.3390/cimb47110940_

Round 1

Reviewer 1 Report

Comments and Suggestions for Authors

The manuscript re-analyses transcriptomic data from mouse models of Chagasic cardiomyopathy (CCC) and ischaemic heart failure (IHF), employing what the authors refer to as the 'Genomic Fabric Paradigm'. While the topic is interesting, particularly the attempt to link mitochondrial transcriptional remodelling to functional recovery following bone marrow cell therapy, I have several concerns about both the conceptual framing and the methodological rigour.

  1. The work relies entirely on existing GEO datasets and there is no integration of proteomic or functional validation. Without any protein-level or physiological corroboration, it is difficult to assess whether the observed transcriptional changes translate into meaningful studies of mitochondrial function in these disease models.
  2. The interpretation of mitochondrial pathways appears to contradict prior literature. For example, numerous studies have reported increased mitochondrial calcium in IHF models (Eur J Heart Fail. 2025 Sep; 27(9): 1720–1736; Basic Res Cardiol. 2024 Jun 19;119(4):569–585). In fact, increased ROS production is often due to increased mitochondrial calcium levels. As the author stated, if all ETCs are downregulated and the calcium level is reduced, what is the major source of ROS production?
  3. The mechanisms of lipotoxicity are rather complicated. It is difficult to conclude that increased fatty acid oxidation is the main cause of lipotoxicity. In fact, reducing FAO can also cause lipotoxicity. (Circ Res. 2016 May 27;118(11):1736–51.)
  4. The small number of replicates (n = 4) and the lack of information about the sex or age of the mice limits the interpretability of the results. Were both models processed under identical conditions?

Based on the above comments, I would suggest rejecting this paper for publication in Current Issues in Molecular Biology.

Author Response

  1. The work relies entirely on existing GEO datasets and there is no integration of proteomic or functional validation. Without any protein-level or physiological corroboration, it is difficult to assess whether the observed transcriptional changes translate into meaningful studies of mitochondrial function in these disease models.

The experiments that generated the data used in this study were performed in 2011 in Chagas Institute from Rio de Janeiro (animal work, pathophysiology, microscopy, proteomics) and in IacobasLab (transcriptomics) located at the time at Albert Einstein College of Medicine from New York. The Brazilian colleagues together with other colleagues from Albert Einstein College of Medicine (our partners in a 10y NIH PO1 grant) published dozens of papers investigating all kinds of pathophysiological, behavioral, microscopical and proteomic aspects of the two cardiomyopathies on the same models. The physiological implications of the mitochondrial damages in both Chagas disease and post ischemic failure have been well documented in the literature and cited by us (refs. 16, 17, 35).

The purpose of the present study was not to duplicate previous work of other groups but to reveal the never before analyzed alterations of the expression control and interplay of the respiratory complexes, and that the cell treatment recovers the gene expression profile but under different transcriptomic organizations. After our knowledge, nobody before us studied the control, the interplay and the remodeling of the respiratory complexes in these two heart conditions.

About the proteomic validation: the amount of transcripts and proteins are not proportional; at low protein level the gene expression is up-regulated, at high protein level the gene expression is down-regulated.  Therefore, a protein-level study would not confirm, nor infirm our transcriptomic findings.

2.  

  1. The interpretation of mitochondrial pathways appears to contradict prior literature. For example, numerous studies have reported increased mitochondrial calcium in IHF models (Eur J Heart Fail. 2025 Sep; 27(9): 1720–1736; Basic Res Cardiol. 2024 Jun 19;119(4):569–585).

It is true that calcium level increases in acute phases of ischemia, but it was reported as lower in post ischemic heart failure (J Physiol. 2017; 595(12):3753-3763. doi: 10.1113/JP273609) that is case in our mouse model.

In fact, increased ROS production is often due to increased mitochondrial calcium levels. As the author stated, if all ETCs are downregulated and the calcium level is reduced, what is the major source of ROS production?

We agreed and removed the sentence about the ROS increase. 

3. The mechanisms of lipotoxicity are rather complicated. It is difficult to conclude that increased fatty acid oxidation is the main cause of lipotoxicity. In fact, reducing FAO can also cause lipotoxicity. (Circ Res. 2016 May 27;118(11):1736–51.)

You are absolutely right, and we have corrected the mistakes both in the figures and the text and added the recommended citation

4. 

  1. The small number of replicates (n = 4)

Regardless of the gene expression platform used, the transcriptomic experiments are affected by ~30-35% technical noise. With Iacobas’ optimized “multiple yellow” protocol, it was reduced to the theoretical limit of ~17-20%. Therefore, more than 4 biological replicas will only increase the cost without increasing accuracy because the potential gain in statistical significance will be “eaten” by the noise. This is one of the reasons why we have replaced the arbitrarily introduced uniform criterion of 1.5x fold-change to consider a gene as significantly regulated with the CUT criterion computed for every single gene to account for both technical noise of the probing microarray spots and biological variability. Thus, we minimized the false positives for the very unstably expressed genes for which the 1,5x criterion is too low and the false negatives for the very stably expressed genes for which the 1.5x criterion was two high. 

the lack of information about the sex or age of the mice limits the interpretability of the results. Were both models processed under identical conditions?

Thank you for signaling this lack of information. Now, we mentioned in the Methods that all mice were adult males from the same strain and processed under identical conditions.

Based on the above comments, I would suggest rejecting this paper for publication in Current Issues in Molecular Biology.

We hope that the revised version responds to your well-justified criticism

Reviewer 2 Report

Comments and Suggestions for Authors

Dumitru A. Iacobas et. al investigated in the study entitled “Mitochondrial collapse responsible for Chagasic and post ischemic heart failure is reversed by cell therapy under different transcriptomic topology” whether there are mitochondria-related genomic alterations in failing hearts due to Chagasic cardiomyopathy and ischemic heart failure before and after stem cell treatment.

Minor comment 1

Please after the second paragraph, you may consider reorganizing your paragraphs to improve contextualization. Please see the detailed suggestion below:

  • Line 56: “…the development of the disease.”
  • Third paragraph (Lines 57–60): currently “Transcriptomic studies…”. I suggest moving this paragraph to follow after Lines 65–73 (Myocardial infarction…).

This way, the content will be reorganized as follows:

Decades after initial infection with the parasitic euglenoid Trypanosoma cruzi [7], 44 transmitted by the so-called “kissing bug” [8], ~30% of individuals can develop chronic 45 Chagasic cardiomyopathy (CCC), a congestive heart failure and dilated cardiomyopathy 46 [9 - 11]. It is estimated that CCC affects about 7 million of people worldwide, most of them 47 in Latin America [12], and became endemic recently even in the United States [13]. Alt-48 hough the pathogenesis of CCC remains a matter of debate [14, 15], the involvement of 49 cardiac mitochondria was first demonstrated by Garg et al. in 2003 [16] and confirmed by 50 Báez et al. in 2011 [17]. They found that a year post-infection parasite persistence and in-51 flammation were associated with structural and functional alterations in mouse cardiac 52 mitochondria in a parasite strain-dependent manner [18]. Common comorbidities include 53 dyslipidemia, hypertension [19] and might lead to cryptogenic stroke [20]. As expected 54 for any infectious disease, CCC triggers the immune response [21] that changes during 55 the development of the disease [22].

Myocardial infarction [32], described as the cardiomyocytes’ death caused by insufficient oxygen supply, whose definition and management are still under debate [33, 34], is directly related to mitochondrial dysfunction [35] and affects several functional pathways [36, 37].

Previous studies on mouse models have shown that myocardial infraction induced by interruption of blood supply [38] activates strong inflammatory response and leads to ventricular remodeling and ischemic heart failure (IHF) [39]. These effects were also reversed by injecting bone marrow-derived mononuclear stem cells into the cardiac scar 71 tissue [40]. Whatever the cause, heart failure has severe consequences on all organs, including triggering various mental disorders [41].

Transcriptomic studies on cardiomyocytes and heart left ventricle of CCC mice revealed remodeling of immune response [23], extracellular matrix, cell adhesion [24], intercellular communication via gap junction channels [25, 26], mitochondrial oxidative phosphorylation [27], JAK/STAT signaling and cell cycle [28] functional pathways.

The progenitor cells are well recognized for their therapeutic potential in cardiac regeneration [29]. We found that injecting bone marrow-derived stem cells restores most of the heart function and recovers most of the normal heart gene expression profile [30, 31].

After this adjustment, please keep the remaining paragraphs in their current order.

Minor Comment 2

Please note that in line 68, the term should be corrected from “myocardial infraction” to “myocardial infarction”.

Minor Comment 3

In the introduction (Line 73), you mention mental disorders. However, this information is not further addressed in the discussion. Therefore, I suggest removing it, since it is not integrated into the main argument of the manuscript.

Minor Comment 4 - Figure 1

For the color code, please do not mention the gene. Instead, specify only the color assigned to each condition: Red – upregulated, Yellow – not regulated, Green – downregulated.

Suggestion: Please specify more clearly which condition is represented in Figure A and Figure B. In addition, it would be better to place Figures A and B side by side, so that the differences can be more easily visualized.

Minor comment 5 - Figure 2

I suggest following the same organization used in Figure 1.

Minor comment 6

Please provide a graphic representing the gene expression levels of chagasic disease and post-ischemic heart failure, showing the fold change of each gene described in the corresponding table. This will make it easier to visualize the differences compared to looking only at the tables. The graphic should be placed close to the corresponding table.

Suggestion: If you still have frozen samples available from this previous work, would be possible to perform a western blot to evaluate the OXPhos protein profile? In this way you may demonstrate the impact on mitochondrial metabolism. In addition, if you have paraffin-embedded samples, you might be able to assess the protein levels of key genes discussed in your work by immunofluorescence (IF). This would further strengthen your work.

Author Response

Minor comment: Please after the second paragraph, you may consider reorganizing your paragraphs to improve contextualization. 

Very good suggestion. We changed the text accordingly

Please note that in line 68, the term should be corrected from “myocardial infraction” to “myocardial infarction”.

 corrected

Minor Comment 3

In the introduction (Line 73), you mention mental disorders. However, this information is not further addressed in the discussion. Therefore, I suggest removing it, since it is not integrated into the main argument of the manuscript.

 Conclusions now mention the link between myocardial infarction and depression and other mental disorders as have recently been addressed by us in a published review

Minor Comment 4 - Figure 1

For the color code, please do not mention the gene. Instead, specify only the color assigned to each condition: Red – upregulated, Yellow – not regulated, Green – downregulated.

We now mention this in the figure legends

Suggestion: Please specify more clearly which condition is represented in Figure A and Figure B. In addition, it would be better to place Figures A and B side by side, so that the differences can be more easily visualized.

 We agree that it would be nice to have all four figures (1ab and 2ab) in 2 x 2 arrangement on one page but it seems difficult to organize the space  

Minor comment 5 - Figure 2

I suggest following the same organization used in Figure 1.

Minor comment 6

Please provide a graphic representing the gene expression levels of chagasic disease and post-ischemic heart failure, showing the fold change of each gene described in the corresponding table. This will make it easier to visualize the differences compared to looking only at the tables. The graphic should be placed close to the corresponding table.

 We accepted your suggestion and added the graphic representation of the expression ratios

Suggestion: If you still have frozen samples available from this previous work, would be possible to perform a western blot to evaluate the OXPhos protein profile? In this way you may demonstrate the impact on mitochondrial metabolism. In addition, if you have paraffin-embedded samples, you might be able to assess the protein levels of key genes discussed in your work by immunofluorescence (IF). This would further strengthen your work.

The experiments that generated the data used in this study were performed in 2011 in Chagas Institute from Rio de Janeiro (animal work, pathophysiology, microscopy, proteomics) and in IacobasLab (transcriptomics) located at the time at Albert Einstein College of Medicine from New York. The Brazilian colleagues quantified several components of the immune response and the heart anatomical changes but not the mitochondria physiology and proteins. When IacobasLab moved to New York Medical College (2013) and then to Prairie View A&M University (2017), most of the frozen samples were discarded, so, although interesting, the proposed studies are no longer possible.